# Targeting Cardiac Metabolism in Heart Failure with PPARα Agonists: A Review of Preclinical and Clinical Evidence

**DOI:** 10.3390/biomedicines13092080

**Published:** 2025-08-26

**Authors:** Carla Handford, Laura Stirling-Barros, Mahboube Ganji-Arjenaki, Masliza Mahmod, Milad Nazarzadeh, Malgorzata Wamil

**Affiliations:** 1Green Templeton College, University of Oxford, Oxford OX2 6HG, UK; carla.handford@gtc.ox.ac.uk (C.H.); laura.stirling-barros@medschool.ox.ac.uk (L.S.-B.); 2Department of Medical Bioinformatics, School of Advanced Technologies, Shahrekord University of Medical Sciences, Shahrekord 9W47+P62, Iran; 3Cardiovascular Science, Perspectum, Oxford OX4 2LL, UK; masliza.mahmod@perspectum.com; 4Deep Medicine, Nuffield Department of Women’s and Reproductive Health, University of Oxford, Oxford OX3 9DU, UK; 5Cardiology Department, Great Western Hospital NHS Trust, Swindon SN3 6BB, UK; 6Cardiology Department, Mayo Clinic Healthcare, London W1B 1PT, UK

**Keywords:** heart failure, fibrates, PPARα agonists, diabetes, fenofibrate, myocardial fibrosis, left ventricular hypertrophy

## Abstract

**Background and objective:** Heart failure (HF) is associated with high morbidity, mortality, and healthcare costs. Its prevalence continues to rise, particularly in the context of ageing populations and increasing rates of metabolic comorbidities such as type 2 diabetes and obesity. We aimed to assess the therapeutic potential of repurposing PPARα agonists for the treatment of HF. **Method:** We conducted a comprehensive literature review to evaluate preclinical and clinical evidence investigating the potential of PPARα agonist drugs in reducing HF. We did not apply any restrictions on the study design. **Results:** The current body of evidence consists of preclinical mechanistic studies, emerging pharmacogenetic data, and post hoc analyses of large randomised clinical trials (RCTs) that included HF endpoints. No dedicated, HF-specific RCTs of PPARα agonists were identified. These studies support the hypothesis that PPARα agonists may link metabolic modulation with cardiac remodelling. Preclinical models demonstrate potential therapeutic benefits, such as enhanced myocardial energy metabolism and attenuation of fibrosis and inflammation, as well as context-dependent risks, including possible deleterious effects in advanced HF or off-target mechanisms. Prior failures of fibrates to improve cardiovascular outcomes in some trials and concerns in PPARα-deficient states underscore the complexity of metabolic therapies in HF. These findings support a more stratified, phenotype-driven approach to therapy. RCTs specifically designed to evaluate HF outcomes are essential to clarify whether PPARα agonists can complement established neurohormonal treatments, particularly in the context of the rising burden of HFpEF associated with obesity and type 2 diabetes. **Conclusions:** PPARα agonists represent a promising class within the emerging therapeutic framework of metabolic heart failure. They are inexpensive, generally well tolerated, and address several pathophysiological mechanisms of HF. Preliminary evidence suggests that fenofibrate may delay or prevent HF in high-risk diabetic populations. However, rigorous, dedicated trials are needed to establish their clinical utility.

## 1. Introduction

The global prevalence of heart failure (HF) is rising, placing an increasing burden on healthcare systems due to escalating treatment costs and resource utilisation [1]. The number of individuals living with HF nearly doubled from 33.5 million in 1990 to 64.3 million by 2017 [1,2]. This surge is mainly attributable to population ageing and the growing incidence of key risk factors for HF preserved ejection fraction (HFpEF), such as type 2 diabetes mellitus (T2DM), hypertension (HTN), and obesity [3]. Cardiovascular complications, including HF, remain among the leading causes of morbidity and mortality in people with diabetes [3]. While HF affects 1–3% of adults in high-income countries, prevalence estimates in low- and middle-income regions remain uncertain due to limited diagnostic infrastructure and data availability [4,5,6]. The economic burden of HF is disproportionately higher in patients with coexisting T2DM, obesity, and HTN, further intensifying the cost pressures on healthcare services [7].

Among the cellular and metabolic derangements implicated in the progression of HF, alterations in myocardial energy metabolism have emerged as a critical pathogenic mechanism [8]. Under physiological conditions, the adult heart derives most of its energy through mitochondrial β-oxidation of long-chain fatty acids. In the failing heart, this metabolic reliance shifts towards increased glucose utilisation—a change that, while initially compensatory, ultimately results in reduced adenosine triphosphate (ATP) generation efficiency (impaired energetics), myocardial steatosis, accumulation of lipotoxic intermediates such as ceramides and diacylglycerols, mitochondrial dysfunction, and progressive contractile impairment [9,10]. This maladaptive reprogramming of cardiac metabolism has prompted growing interest in pharmacological strategies to restore fatty acid oxidation (FAO) to preserve myocardial function [8].

Peroxisome proliferator-activated receptor alpha (PPARα) is a ligand-activated nuclear receptor that regulates genes’ transcription in FAO, lipid transport, and mitochondrial biogenesis [11]. In preclinical models, activation of PPARα has been shown to enhance mitochondrial efficiency, reduce myocardial steatosis, suppress pro-inflammatory signalling, and attenuate cardiac fibrosis [12]. Several pharmacological agents from the fibrate class (e.g., fenofibrate, bezafibrate, clofibrate) have demonstrated the ability to modulate these pathways. Furthermore, compounds such as Astragaloside IV, a natural activator of PPARα, have shown promise in experimental HF models by promoting metabolic shifts from glycolysis to FAO, improving cardiac energetics, and reducing pathological remodelling [13].

Despite these encouraging mechanistic insights, clinical translation remains limited. While fibrates are widely used for treating dyslipidaemia and atherogenic metabolic profiles, no randomised controlled trials (RCTs) have formally assessed PPARα agonists for the prevention or treatment of HF as a primary outcome. Existing clinical data, derived largely from post hoc subgroup analyses of lipid-lowering trials, provide suggestive but inconclusive evidence of benefit, particularly in patients with diabetes or metabolic syndrome. Concerns regarding adverse effects, including myopathy and renal function deterioration, have further attenuated initial enthusiasm and underscored the need for a more nuanced understanding of risk-benefit profiles in this population [14].

We aimed to review and appraise the existing body of evidence on the potential role of PPARα agonists in HF management. We explore the molecular mechanisms by which PPARα activation may modulate cardiac metabolism, inflammation, and structural remodelling [11,15] by evaluating data from preclinical and translational studies, and review the limited but emerging randomised clinical trial literature.

## 2. Methods

### 2.1. Search Strategy

Comprehensive literature searches were performed using the Ovid Medline and EMBASE databases and ClinicalTrials.gov to identify relevant preclinical, translational, and randomised clinical trials. Additional searches were conducted via PubMed, Scopus, and Web of Science to ensure broad coverage. Keywords and Medical Subject Headings (MeSH) included: “PPARα”, “peroxisome proliferator-activated receptor alpha”, “fenofibrate”, “clofibrate”, “bezafibrate”, “heart failure”, “cardiac metabolism”, “fatty acid oxidation”, “fibrosis”, “inflammation”, and “oxidative stress”. The search strategy for Clincialtrials.gov included “HF: as a condition” and “fibrate” or “fenofibrate or gemfibrozil or clofibrate” as other terms. See Appendix A for details.

### 2.2. Inclusion and Exclusion Criteria

Studies were eligible for inclusion if they met the following criteria: investigated the effects of PPARα activation or pharmacological agonists (e.g., fenofibrate, clofibrate, bezafibrate) on cardiac structure or function, reported outcomes related to myocardial energy metabolism, inflammation, fibrosis, oxidative stress, or HF-specific clinical endpoints, or were original research articles involving animal models or genetic models (e.g., PPARα knockout or overexpression). The exclusion criteria were non-cardiac applications of PPARα agonists and studies where PPARα activation was not the primary intervention.

### 2.3. Data Extraction and Synthesis

Two authors (C.H. and L.S.B.) independently reviewed eligible studies and extracted data into structured templates. Full-text articles were retrieved for studies deemed potentially eligible. Disagreements were resolved through discussion or adjudication by a third reviewer (M.W.). Additionally, M.W. reviewed RCTs for inclusion in the analysis. Extracted information included study design, model system, sample size, type and dose of PPARα agonist, mechanistic focus (e.g., fatty acid oxidation, fibrosis, inflammation), and reported outcomes. Data were categorised into three domains: (a) genetic models involving PPARα overexpression or gene deletion, (b) preclinical pharmacological models—studies using PPARα agonists, (c) clinical studies including RCTs and HF endpoints involving fibrate therapy.

Given the heterogeneity in study design and outcomes, a qualitative narrative synthesis was conducted. Quantitative meta-analysis was not performed due to methodological and population differences among the included studies.

### 2.4. Findings

#### 2.4.1. Study Flow

Figure 1 presents a PRISMA flow diagram for preclinical studies summarising study identification and selection. A selection of RCTs was carried out separately as described in the Appendix A.

#### 2.4.2. Study Characteristics

A total of 19 preclinical studies met inclusion criteria and were categorised into two major groups based on experimental design. Table 1 summarises 15 pharmacological studies evaluating the effects of PPARα agonists in various animal models of HF. These studies report on a range of therapeutic agents (e.g., fenofibrate, clofibrate, WY-14643) and investigate downstream molecular pathways related to fatty acid oxidation (FAO), inflammation, and myocardial fibrosis.

Table 2 summarises four genetic studies employing knockout or transgenic models to examine the mechanistic role of PPARα in cardiac pathophysiology. These studies explored gene-dosage effects, tissue-specific deletions, and pathway perturbations linked to energy metabolism and structural remodelling in HF.

Outcomes of interest across all studies included measures of left ventricular function (ejection fraction, fractional shortening), FAO biomarkers (CPT1, acylcarnitines), pro-inflammatory cytokines (TNF-α, IL-6), fibrosis-related proteins (TGF-β, collagen types I and III), and in clinical trials, HF hospitalisation. Each study’s methodological limitations are documented and summarised in Appendix A, including issues related to population heterogeneity, treatment duration, and endpoint definitions.

Given the substantial heterogeneity in study models, design, dosing regimens, and outcome assessment across both pharmacological and genetic studies, we have synthesised findings narratively to ensure contextual interpretation.

Table 3 presents three large-scale RCTs evaluating fibrates in patients with T2DM and dyslipidaemia, focusing on cardiovascular outcomes, but also specifically reporting HF outcomes. Two of these trials, the Veterans Affairs HDL Intervention Trial (VA-HIT) [16] and the ACCORD-Lipid trial [17], reported hazard ratios (HRs) for HF hospitalisations or events, providing the most direct evidence to date regarding the potential role of PPARα agonists in HF prevention.

**Table 1 biomedicines-13-02080-t001:** Overview of the pharmacological studies identified through a systematic review, highlighting the therapeutic agents used in the study, associated molecular pathways, and their reported effects on heart failure-related endpoints.

Author	Animal Model	Heart Failure Model	Sample Size	Drug Regime	Drug Control	Mechanism	Beneficial or Harmful	Myocardial Fibrosis	Altered Lipid Metabolism	Cardiac Insulin Resistance	Altered Gene Expression	Diastolic Dysfunction	Altered Mitochondrial Homeostasis	Cardiac Inflammation	Brief Description
Young et al., 2001 [18]	M SD 200–255 g rats and mice PPARα −/−	Cardiac workload was either increased (pressure overload by AC) or decreased (mechanical unloading by heterotopic transplantation)	*n* = 5–10 observations	WY-14643 (0.01% *w*/*w*) was added to standard powdered Purina rodent chow fed to rats for 4 days	Regular chow diet	Altered mitochondrial homeostasis	B	N/A	Y	N/A	Y	Y	Y	N/A	UCP3 expression is regulated by PPARα, proposed to have an antioxidant role.
Young et al., 2001 [19]	M SD 225 g rats	Pressure overload (ascending AC for 7 days) resulted in cardiac hypertrophy	*n* = 9–13 observations	WY-14643 (0.01% *w*/*w*) in powdered chow	Regular chow diet	Diastolic dysfunction	H	Y	N/A	N/A	Y	Y	N/A	N/A	Reactivation of PPARα in the hypertrophied heart is linked to contractile dysfunction.
Aasum et al., 2003 [20]	F db/db 6–11 weeks mice	db/db mice (non-diabetic littermate controls)	Non-diabetic *n* = 10, db/db untreated *n* = 23, db/db treated *n* = 27	At 8 weeks given BM 17.0744 for 4–5 week (dose calculated from daily water intake between 24.5 +/− 1.35 and 37.9 +/− 2.5 mg/kg/day)	No drug	Altered lipid metabolism	B	N/A	Y	Y	N/A	N/A	N/A	N/A	Chronic PPARα agonist treatment reduced fatty acid oxidation and increased glycolysis and glucose oxidation, correcting diabetes-induced abnormalities. There was no improvement in LV contractile function.
Ichihara et al., 2006 [21]	M DS 7–18 weeks rats	Heart failure secondary to chronic hypertension induced by salt sensitivity	Low salt = *n* = 12, high salt + vehicle *n* = 24, high salt + fenofibrate 30 mg/kg *n*= 12, high salt + fenofibrate 50 mg/kg *n* = 12	Two groups: low dose of fenofibrate (30 mg/kg/day), and high dose of fenofibrate (50 mg/kg/day) administered orally by gastric gavage once daily from 7–18 weeks of age	DS rats maintained on a diet of 0.3% NaCl until 18 weeks and 3% gum arabic vehicle	Reduced cardiac inflammation and fibrosis	B	Y	N/A	N/A	Y	Y	N	Y	Fenofibrate reduced cardiac hypertrophy, inflammation, fibrosis, diastolic and systolic dysfunction.
Morgan et al., 2006 [22]	M Wistar-Kyoto 8–20-week 250 g rats	Heart failure was induced by coronary artery ligation	Infarct-induced HF *n* = 38, sham operated *n* = 10, untreated *n* = 10, high fat *n* = 15, fenofibrate *n* = 13	Fenofibrate (150 mg/kg, milled into the food)	Regular chow diet	Altered lipid metabolism	B	N/A	Y	N/A	Y	Y	Y	N/A	Prolonged administration of a PPARα agonist increases fatty acid oxidation capacity, although this beneficial effect can be reduced by a high-fat diet.
Pruimboom Brees et al., 2006 [23]	WT and PPAR-null 9–12-week mice on an SV 129 background	None	Each group *n* = 8	PPAR- fibric acid derivative (most selective for PPARα)	Water (vehicle)	Increased cardiac oxidative stress and necrosis	H	Y	Y	N/A	Y	N/A	Y	N/A	Activation of PPARα leads to increased cardiac fatty acid oxidation and oxidative stress intermediates resulting in cardiomyocyte necrosis.
King et al., 2007 [24]	M Wistar < 16 weeks rats	None	Diabetes *n* = 9, control *n* = 8, standard diet *n* = 12, fenofibrate *n* = 12, untreated or treated with fenofibrate *n* = 9	Fenofibrate (300 mg/kg/day in the chow) for 4 weeks	Regular chow diet	Mitochondrial homeostasis	B	N/A	Y	N/A	Y	N/A	Y	N/A	PPARα agonist upregulates MTE1 to regulate fatty acid accumulation in the mitochondrial matrix when the heart is exposed to elevated levels.
Anne D. Hafstad et al., 2009 [25]	M BALB/cA Bom 12 weeks mice	40 min low-flow ischaemia followed by 35 min reperfusion (5 min in Langendorff and 30 min in working mode)	*n* = 22–25 mice in total	Mice treated with TTA (0.5% *w*/*w*) for 8 days	Age matched untreated	Altered lipid metabolism	H	N/A	R	N/A	Y	Y	N/A	N/A	In vivo administration of synthetic PPARα ligand increased FAO and decreased glucose oxidation, associated with decreased cardiac efficiency and reduced post-ischaemic functional recovery.
Chen et al., 2010 [26]	F Diabetic KKAy 38–42 g mice	Inoculated with encephalomyocarditis virus	Vehicle group *n* = 45, early agonist group *n* = 39, late group agonist *n* = 18	2 groups: WY-14643-early—received daily dose 50 mg/kg starting 3 days before viral inoculation, WY1463-late—received simultaneously at viral inoculation	Vehicle (dimethyl sulfoxide)	Reduced cardiac inflammation	B	Y	Y	N/A	Y	N/A	Y	Y	PPARα agonist was cardioprotective, perhaps due to reduced inflammation.
Haemmerle et al., 2011 [27]	M+F Atlg KO < 10 weeks mice	None	*n* = 4 mRNA Glycogen content *n* = 9 in each group Mitochondria analysis *n* = 4–6 in each group Oxygen consumption *n* = 6 KO *n* = 5–7 depending on experiment	WY-14643 (0.1% *w*/*w*) for time indicated and fenofibrate (0.2% *w*/*w*) for 10 weeks provided to separate mice via feeding chow diet	Regular chow diet	Altered gene expression	B	N	R	N/A	N/A	N	N/A	N/A	PPARα agonists in mice with decreased PPARα target gene expression reverse mitochondrial defects, restore normal heart function, and prevent premature death.
Jia et al., 2014 [28]	M 10 week 24–26 g mice	Cardiac hypertrophy induced by thoracic transverse AC	*n* = 3–5 for each group in experiments	Fenofibrate (50 g/kg) via gavage after operation for 4 weeks	Saline	Reduced cardiac inflammation	B	N/A	N/A	N/A	Y	Y	N/A	Y	Fenofibrate modulates basal and lipopolysaccharide (LPS)-stimulated HMGB1 expression and secretion in cardiomyocytes. Fenofibrate also prevents cardiac hypertrophy.
Ibarra-Lara et al., 2016 [29]	M Wistar 24 weeks 50 g rats	MetS rats (received 30% sugar in drinking water for 24 weeks) and controls (tap water for drinking). MI achieved by occluding LAD coronary artery for 60 min (sham control)	*n* = 5 per group for experiments	At end of 24 weeks, given 2-week oral treatment of fenofibrate (100 mg/kg/day)	Vehicle treatment (NaCl 0.9%)	Reduced insulin resistance	B	R	R	N	Y	N/A	N/A	Y	Fenofibrate reduced triglycerides, non-HDL cholesterol, insulin levels, and insulin resistance index in MetS animals, whilst promoting an antioxidant environment.
Kaimoto et al., 2017 [12]	M 10-week mice	Pressure-overload heart failure model in mice through transverse AC. Transgenic PPARα overexpression induction using Tet-Off system	*n* = 6–10 per group for different experiments	WY-14643 (0.01% *w*/*w*) in powdered chow. Operation at 10 weeks. At 14–18 week of age, powdered diet with WY-14643	Regular chow diet	Altered lipid metabolism	B	Y	Y	N/A	Y	Y	N/A	N/A	PPARα activation during pressure-overloaded heart failure improved myocardial function and energetics.
Ibarra-Lara et al., 2019 [30]	M Wistar 300–350 g rats	LAD coronary artery ligation (MI) (sham control)	*n* = 6 per group for experiments	7 days post-MI, animals were given clofibrate (100 mg/kg) for 7 days	Vehicle (vegetable oil) by intraperitoneal injection	Reduced cardiac inflammation	B	R	N/A	N/A	N/A	Y	N/A	R	Clofibrate decreases late inflammation and partially reverses LV remodelling and functional damage.
Sanchez-Aguilar et al., 2023 [31]	M Wistar 24 weeks rats	MetS rats received 30% sucrose in drinking water for 24 weeks, whereas controls given tap water. LAD coronary artery ligation gave ischaemic reperfusion model (sham control)	*n* = 5–6 per group for experiment	At 24 weeks, MetS animals given clofibrate (100 mg/kg/day) by intraperitoneal injection for 7 days (pre-treatment to I/R model)	Control vehicle injection	Reduced cardiac inflammation	B	R	R	N	Y	N/A	N/A	R	Pre-treatment with clofibrate decreased cardiac inflammation, reduced myocardial fibrosis and apoptosis, whilst improving insulin sensitivity.

Abbreviations: M—male, F—female, WT—wild type, DS—Dahl Sensitive, SD—Sprague-Dawley, MetS—Metabolic Syndrome, AC—aorta constriction, LAD—left anterior descending, B—beneficial, H—harmful, Fenofibrate, clofibrate, BM 17.0744, WY-14643—specific PPARα agonists, Y—yes, N—no, N/A—not applicable, R—reduced, LV—left ventricular.

**Table 2 biomedicines-13-02080-t002:** Overview of the genetic studies identified through the systematic literature review, highlighting the models used, molecular pathways examined, and their relevance to heart failure pathophysiology.

Author	Animal Model	Heart Failure Model	Sample Size	Mechanism	Beneficial or Harmful	Myocardial Fibrosis	Altered Lipid Metabolism	Cardiac Insulin Resistance	Altered Gene Expression	Diastolic Dysfunctions	Altered Mitochondrial Homeostasis	Cardiac Inflammation	Brief Description
Finck et al., 2002 [32]	M 8–16 weeks 22–30 g mice, some db/db MHC-PPARα mice	Diabetes induced by single intraperitoneal injection of STZ or db/db mice (control—vehicle injection). MHC-PPARα mice (control—non-transgenic littermates).	*n* = 3–7 per group for experiments	Altered lipid metabolism	H	N/A	Y	Y	Y	Y	Y	N	MHC-PPARα heart shows similar expression of genes involved in myocardial fatty acid utilisation and reciprocal downregulation of myocardial glucose pathways to the diabetic heart, ultimately altering cardiac myocyte lipid balance and resulting in hypertrophy and ventricular dysfunction.
Park et al., 2005 [33]	M MHC-PPARα 2–14 weeks 27 g mice	MHC-PPARα overexpression (age-matched WT control).	*n* = 4–6 for most experiments *n* = 6–13 for hyperinsulinaemic experiments	Cardiac insulin resistance	H	N/A	N	Y	Y	Y	N/A	N/A	Increased activity of PPARα results in insulin resistance and defects in insulin signalling and STAT3 activity, reducing cardiac function.
Marionneau et al., 2008 [34]	M + F MHC-PPARα 5–6 weeks mice	MHC-PPARα cardiac-specific overexpression (age-matched WT control).	*n* = 9 in wild type mice group *n* = 9 in MHC-PPARα C57BL/6 mice group	Ventricular Kv current remodelling	Unclear	N/A	Y	N/A	Y	N/A	N/A	N/A	Cardiac-specific activation of PPARα results in ventricular Kv current remodelling in left ventricles, although this is age-related.
Duerr et al., 2014 [35]	M+F MHC-PPARα and C57BL/6J WT 10–12 weeks 20–25 g mice	LAD coronary artery ligation and MHC-PPARα cardiac-specific overexpression,	*n* = 8 in MHC-PPARα group *n* = 6 in wild type group	Myocardial fibrosis	H	Y	N/A	N/A	Y	Y	Y	Y	Cardiomyocyte-specific PPARα overexpression resulted in cardiomyocyte loss and reduced ventricular function. There was increased glycogen deposition, apoptosis, reduced antioxidative capacity, resulting in post-ischaemic inflammation and remodelling.

Abbreviations: MHC—Myosin Heavy Chain, PARα—Peroxisome Proliferator-Activated Receptor Alpha, STZ—Streptozotocin, db/db mice—Diabetic mice homozygous for the leptin receptor mutation, WT—Wild Type, Kv current—Voltage-Gated Potassium Current, LAD—Left Anterior Descending (coronary artery), N/A—Not Applicable, H—harmful, N—no, Y—yes.

**Table 3 biomedicines-13-02080-t003:** Randomised controlled trials investigating the role of fibrate agents on heart failure events.

Trial Name	Ref	Participants	Intervention	Comparison	Primary Outcomes	Heart Failure HR	Trial Design
Veterans Administration Cooperative Study of Atherosclerosis (VA CO-OP) (1973)	[36]	380 men with previous history of stroke	Clofibrate	Placebo	Recurrent strokes	HR not reported; number of events in P/C arms: 4/15	Parallel 1:1
Veterans Affairs HDL Intervention Trial (VA-HIT) (1999)	[16]	2531 men < 75 y old with CVD and dyslipidaemia	Gemfibrozil	Placebo	Composite of myocardial infarction or CVD	0.78 (0.61–0.99); *p* = 0.04	Parallel 1:1
Action to Control Cardiovascular Risk in Diabetes Lipid trial (ACCORD-Lipid) (2010)	[17]	5518 Type 2 diabetes patients with dyslipidaemia and CVD or CV risk factors	Fenofibrate + statins	Statins	MACEs	0.82 (0.65–1.05); *p* = 0.1	Factorial (intensive vs. standard glucose control)

Abbreviations: CVD—cardiovascular disease, MACEs—major adverse cardiovascular event.

Preclinical studies varied widely in design, species, and endpoints, while RCTs such as VA-HIT and ACCORD-Lipid did not prespecify HF outcomes and reflect older treatment eras. Following Cochrane guidance, we therefore applied a structured narrative synthesis rather than meta-analysis, emphasising mechanistic insights while acknowledging the lower evidential weight for causal inference.

### 2.5. Preclinical Studies

#### 2.5.1. The Role of PPARα in Metabolic Modulation and Energy Homeostasis

PPARα is a nuclear receptor crucial for myocardial energy metabolism, particularly in regulating FAO [23]. The heart heavily relies on FAO to meet its considerable energy demands, and dysregulation of this pathway is linked to the onset of HF [8]. When activated, PPARα enhances the expression of genes involved in FAO, such as carnitine palmitoyltransferase 1 (CPT1) and medium-chain acyl-CoA dehydrogenase (MCAD), resulting in increased myocardial ATP production. Aasum et al. [20], using a db/db mouse model, demonstrated that PPARα activation enhanced FAO by regulating gene expression. This, in turn, improved mitochondrial function, often compromised in diabetes, ultimately increasing cardiac metabolic efficiency and improving left ventricular function.

Additionally, the treatment with PPARα activators has also been shown to enhance the activity of mitochondrial thioesterase 1 (MTE1) [24], which is involved in the metabolism of long-chain acyl-CoA molecules, in both diabetic and non-diabetic rat hearts, possibly representing a compensatory mechanism for managing higher lipid loads in the mitochondria. Furthermore, Young et al. found that the expression of uncoupling protein 3 (UCP3) is regulated by PPARα, which is induced under conditions that elevate FAO, further supporting the role of PPARα in regulating mitochondrial FAO [18]. Hence, the disruption of that pathway may affect cardiac efficiency, thermogenesis, and oxidative stress, all of which are critical processes in HF development.

In addition to its well-established role in lipid metabolism, activation of PPARα may also contribute to enhanced insulin sensitivity. In experimental models of metabolic syndrome, fenofibrate administration has been associated with a reduction in myocardial insulin resistance and improved cardiac tissue integrity [29]. This may occur through the modulation of the angiotensin II pathway, which involves decreased expression of angiotensin II receptors and signalling, although further validation is required. As a result, PPARα activation during the early stages of HF development has been proposed to help preserve cardiac output and improve function by increasing FAO and ATP production [37].

Despite encouraging preclinical data, the long-term effects of PPARα activation in the context of HF remain subject to ongoing debate. The heterogeneity of outcomes observed across experimental models (presented in Figure 1 and discussed below) underscores the need for a more nuanced understanding of context-dependent responses, including disease stage, comorbid conditions, and molecular milieu. Further investigation is required to delineate the conditions under which PPARα modulation yields beneficial versus neutral or adverse effects.

Morgan et al. demonstrated that chronic PPARα activation improved cardiac function in a rat model of myocardial ischaemia-induced HF by upregulating the expression of key FAO genes [22]. Still, these cardioprotective effects were attenuated by a high-fat diet, suggesting that dietary factors may modulate the therapeutic efficacy of PPARα agonists [22]. Further studies are required to understand the interaction between PPARα activation and metabolic substrates and their implications for disease progression and treatment outcomes in HF in individuals with diabetes and obesity.

#### 2.5.2. Potential Adverse Effects in Ischaemic Conditions

Evidence suggests that PPARα activation may exert detrimental metabolic effects under certain pathological conditions. Hafstad et al. reported that following ischaemic injury, enhanced FAO driven by PPARα activation reduced mitochondrial and cardiac efficiency, ultimately impairing post-ischaemic functional recovery [25]. This finding challenges the assumption that PPARα activation is uniformly beneficial, highlighting the context-dependent nature of its metabolic effects. While PPARα stimulation optimises lipid metabolism under physiological conditions, its activation in ischaemic states, where efficient ATP production is critical, may exacerbate metabolic inefficiencies and compromise cardiac function. These findings highlight the importance of a context-specific therapeutic strategy, emphasising that PPARα-targeted interventions should be implemented with caution and appropriate regulatory oversight, particularly in patients with established cardiovascular disease or coexisting metabolic disorders such as obesity, to minimise the risk of adverse outcomes.

#### 2.5.3. Anti-Inflammatory and Antioxidant Effects of PPARα Activation

In the present review, three studies reported the effects of PPAR activation on anti-inflammatory and antioxidant properties [26,30,38]. Chronic low-grade inflammation is a defining characteristic of HF, contributing to disease progression through sustained activation of pro-inflammatory signalling pathways [38], and it has been proposed that PPARα activation may exert anti-inflammatory effects, potentially offering a therapeutic role in preventing the development of HF.

For example, in a viral myocarditis model of diabetic mice, PPARα agonists downregulated TNF-α expression and upregulated adiponectin, suggesting a dual role in mitigating inflammation and promoting metabolic homeostasis [26]. Similarly, in a myocardial infarction model, administration of the PPARα agonist clofibrate resulted in a marked reduction in inflammatory biomarkers (ICAM-1, VCAM-1, NF-κB, and iNOS), while enhancing anti-apoptotic pathways, which translated into improved cardiac function, reduced left ventricular dilation, and preserved myocardial contractility [30].

Based on these findings, further evidence suggests that PPARα activation may exert immunomodulatory effects in the context of preexisting cardiovascular pathology, notably by reducing the expression of redox-regulated transcription factors induced by a high-salt diet [21]. Additionally, it decreased T lymphocyte and macrophage infiltration into the left ventricle, along with a reduction in plasma C-reactive protein (CRP) concentrations, indicating systemic anti-inflammatory effects. These findings suggest a potential role for PPARα activation in mitigating cardiac inflammation, thereby preventing left ventricular hypertrophy and systolic dysfunction, key drivers of HF progression.

Beyond inflammation, recent studies have highlighted that PPARα regulates autophagy, a crucial process in maintaining cardiomyocyte homeostasis. In pressure overload-induced HF models, fenofibrate enhanced AMP-activated protein kinase (AMPK) activation, promoting autophagic clearance of damaged organelles and preventing maladaptive hypertrophy [39]. Additionally, it reduced endoplasmic reticulum stress markers (GRP78, CHOP), further protecting against cardiac dysfunction [39]. These findings highlight the potential of fenofibrate as a therapeutic strategy in managing cardiac hypertrophy and dysfunction.

Supporting this, the study by Jie et al. demonstrated that fenofibrate downregulated HMGB1 expression and secretion, reducing inflammation in hypertrophied hearts, improving cardiac function and preventing progression to HF [28]. Similarly, Sanchez-Aguilar et al. suggested that clofibrate alleviated ischaemia/reperfusion injury in rats with metabolic syndrome by modulating the atrial natriuretic peptide (ANP), a compensatory response that promotes improved heart function after injury [31]. However, some discrepancies remain. A separate study reported that chronic treatment with PPARα agonists induced oxidative stress and elevated peroxisomal FAO biomarkers, ultimately leading to cardiomyocyte necrosis [23], implying that time-dependent impacts of this signalling pathway must be better explored.

While these findings highlight potential cardioprotective effects, they underscore the complexity of PPARα-driven immune modulation. While beneficial under certain conditions, suppressing inflammatory pathways may lead to unintended consequences, such as disrupting homeostatic inflammatory signalling. Since inflammation plays a dual role in cardiac remodelling as a pathological driver and a crucial component of tissue repair, indiscriminate PPARα activation could inadvertently compromise adaptive immune processes, particularly in ischaemic or viral-induced myocardial injury. Future research should focus on delineating the delicate balance between beneficial and detrimental inflammatory modulation, ensuring that targeted interventions do not exacerbate metabolic inefficiencies or maladaptive immune responses in patients with HF.

#### 2.5.4. Attenuation of Myocardial Fibrosis as a Target for HF Prevention

Myocardial fibrosis is a key pathological process in the progression of HF, contributing to ventricular stiffness, impaired relaxation, and adverse remodelling [38]. Excessive deposition of extracellular matrix (ECM) proteins, particularly collagen types I and III, disrupts the normal myocardial architecture, reducing compliance and leading to diastolic and systolic dysfunction [40]. Animal studies have demonstrated that fenofibrate can inhibit left ventricular hypertrophy, resulting in myocardial fibrosis [41,42]. Given the pivotal role of fibrosis in HF pathophysiology, targeting fibrotic remodelling through antifibrotic interventions, such as PPARα pathway activation, represents a promising therapeutic strategy in HF management.

Evidence from animal studies suggests that PPARα agonists may exert antifibrotic effects by modulating key profibrotic signalling pathways. Specifically, PPARα activation has been shown to inhibit transforming growth factor-beta (TGF-β) and Smad-dependent fibrotic signalling, thereby attenuating collagen deposition in failing hearts [43]. In models of diabetic cardiomyopathy, fenofibrate reduced interstitial fibrosis and preserved left ventricular compliance, suggesting a protective role in preventing HF development [21]. Beyond its impact on ECM remodelling, PPARα activation has also been implicated in mitigating ischaemic injury. Experimental myocardial infarction models have demonstrated that fenofibrate decreased infarct size. This was secondary to the reduction of apoptosis via upregulation of the anti-apoptotic protein Bcl-2, leading to improved post-ischaemic left ventricular function [30].

Notably, studies utilising genetic models involving cardiac-specific overexpression of PPARα have reported detrimental effects associated with excessive signalling (Table 2). These models exhibit pathophysiological features reminiscent of the diabetic heart [32], including disruptions in cardiac myocyte lipid homeostasis, which results in ventricular dysfunction. Additionally, there is evidence suggesting that chronic PPARα activation in these transgenic models induces systemic insulin resistance [33], compromises antioxidative capacity, and exacerbates post-ischaemic inflammation [35], thereby promoting maladaptive cardiac remodelling and inflammation.

However, while these findings underscore the potential risks of unregulated PPARα activity, it is important to acknowledge the limitations of such transgenic models. The constitutive overexpression of PPARα may represent an artificial system that does not accurately reflect physiological regulation in humans, particularly in the context of metabolic disease. As a result, caution is warranted when extrapolating these findings to clinical scenarios. Instead, a more nuanced approach is required, integrating physiological models that better capture the dynamic regulation of PPARα in disease states. Further research is needed to delineate the precise molecular pathways through which PPARα influences cardiac metabolism and remodelling, ensuring that potential therapeutic strategies effectively balance metabolic benefits with the risk of exacerbating maladaptive processes.

Complementary evidence further supports the cardioprotective role of PPARα activation in fibrosis-related HF progression. Studies indicate that PPARα agonists enhance mitochondrial biogenesis and improve endothelial function, processes which are critical in maintaining myocardial energy homeostasis and vascular integrity [44]. Additionally, PPARα agonists have the ability to suppress oxidative stress (a major driver of fibroblast activation), suggesting that they may confer broader protective effects against adverse cardiac remodelling. However, the precise context in which PPARα activation exerts these benefits remains an area of ongoing debate. A more precise understanding of these pathways may unlock novel therapeutic strategies aimed at halting or even reversing myocardial fibrosis before HF becomes irreversible.

#### 2.5.5. PPARα Agonists as a Drug Target for HF Prevention

Given the substantial preclinical evidence supporting the cardioprotective effects of PPARα agonists (Figure 2), these agents have emerged as promising therapeutic candidates for managing HF. Their capability to restore myocardial energy metabolism, reduce inflammation, and alleviate fibrosis indicates a potential role in complementing established guideline-directed HF therapy, particularly in patients with metabolic-driven cardiomyopathies such as diabetic cardiomyopathy. By addressing myocardial metabolic dysfunction, a process not directly targeted by conventional HF therapies, PPARα agonists could function as adjunctive treatments to optimise cardiac function and slow disease progression.

Importantly, fenofibrate and newer selective PPARα agonists offer mechanistic advantages that may enhance current HF treatment paradigms. Their potential to modulate lipid metabolism, suppress oxidative stress, and preserve mitochondrial integrity suggests that early therapeutic intervention could yield benefits beyond those achieved with standard pharmacotherapy.

However, despite compelling experimental data, the clinical translation of PPARα-targeted therapies remains an area of active investigation, with challenges arising due to inconsistencies in patient responses, likely reflecting the complex interplay between metabolic, inflammatory, and fibrotic pathways in HF. Before these agents can be integrated into routine clinical practice, robust evidence from RCTs is required to confirm their efficacy, safety, and optimal patient selection to identify patient subgroups most likely to benefit from PPARα-targeted therapies and to clarify the long-term effects of PPARα activation in HF prevention. Such stratification may involve genetic markers (e.g., PPARα polymorphisms), comorbidities such as diabetes or obesity, and specific HF phenotypes like HFpEF.

The following section will critically examine the current landscape of clinical trials evaluating PPARα agonists in HF, assessing their translational potential and the challenges associated with their clinical implementation.

#### 2.5.6. Randomised Controlled Trials with Fibrates

Fenofibrate was approved by the Food and Drug Administration (FDA) in 2004 for use in adults as an adjunct to diet for treating mixed dyslipidaemias. It is also licensed in the UK and the European Union. Fenofibrate is primarily indicated for treating hypertriglyceridaemia and mixed dyslipidaemia, conditions commonly associated with abnormal lipid profiles in patients with T2DM.

Several large-scale RCTs investigated the effects of fenofibrate on cardiovascular outcomes and reducing cardiovascular events in patients with T2DM and dyslipidaemia, but the results have been mixed and inconsistent [14,17,45]. A recently published Lowering Events in Non-proliferative Retinopathy in Scotland (LENS) trial demonstrated that fenofibrate effectively slows the progression of diabetic retinopathy. Significantly, this benefit appears independent of its lipid-lowering effects, suggesting that fenofibrate may have a broader therapeutic role in managing diabetes-related complications beyond dyslipidaemia [46].

Cardiovascular trials of fibrates have primarily focused on major adverse cardiovascular events (MACEs), including cardiovascular death, nonfatal myocardial infarction (MI), and stroke endpoints, which were critical in evaluating the efficacy of fibrate therapy in reducing the risk of significant cardiovascular incidents in patients with metabolic disorders such as diabetes and dyslipidaemia. Meta-analyses have shown a 10% risk reduction for MACEs, with a larger benefit (∼35% risk reduction [95% CI 22–64]) in individuals with atherogenic dyslipidaemia (low HDL cholesterol and high triglycerides) [47] or a common PPARα polymorphism (∼49% risk reduction [95% CI 28–66]) [48]. A large meta-analysis of RCTs of fibrates, as a class of lipid-lowering medications, collated data on HF events reported in three trials involving 8581 participants and 584 HF events (the Action to Control Cardiovascular Risk in Diabetes Lipid trial (ACCORD-Lipid) [17] with fenofibrate, the Veterans Affairs HDL Intervention Trial (VA-HIT) testing gemfibrozil [16], and the Veterans Administration Cooperative Study of Atherosclerosis (VA CO-OP) testing clofibrate (Table 3) [45,47]. It concluded that fibrate therapy did not reduce HF hospitalisation. However, significant heterogeneity in the results was noted, primarily due to the VA CO-OP Atherosclerosis trial (RR 0.94 (0.65–1.37), I^2^ 72.6%, *p* for heterogeneity 0.026).

The post hoc analysis of the ACCORD-Lipid trial showed that the treatment with fenofibrate versus placebo in addition to statins reduced the risk of HF hospitalisation and cardiovascular death by 18% (95% CI 0–32; *p* = 0.048) [49]. This trial allowed for the assessment of lipid-lowering strategies alongside glucose and blood-pressure control interventions, enabling complex interaction analyses. Participants in the trial were randomly assigned to different combinations of interventions to control glucose, lipids, and blood pressure. While this structure allowed simultaneous testing of multiple treatments, it also introduced potential interactions between them. The effect was larger in the standard glucose control group, with a 36% risk reduction of HF hospitalisation and cardiovascular death and 40% of HF hospitalisation alone [49]. No benefits were observed among those on intensive glycaemic control (target HbA1c < 6%), implying that the complex design might have influenced the overall result. This finding suggests that the intensity of glucose control may have modified the response to lipid-lowering therapy. It underscores the importance of interpreting results within the context of factorial trial designs, where interactions between interventions can significantly influence outcomes.

#### 2.5.7. Limitations of Existing Clinical Evidence and the Need for HF-Specific Trials

Despite compelling preclinical evidence suggesting a cardioprotective role for PPARα agonists, translating these findings into clinical practice remains limited due to a lack of dedicated RCTs evaluating HF as a prespecified primary endpoint. While fenofibrate has demonstrated efficacy in treating dyslipidaemia and reducing cardiovascular risk, its impact on HF prevention or treatment has not been systematically assessed. Previous studies have not shown consistently HF-specific benefits, owing to several methodological and design limitations. First, substantial heterogeneity in study populations might have obscured therapeutic effects in clinically relevant subgroups, such as individuals with metabolic syndrome or HF with preserved ejection fraction. Second, many trials have employed composite cardiovascular endpoints such as MACEs, which may dilute or mask potential HF-specific effects. The most recent RCT (ACCORD-Lipid) concluded in 2010, while others date back over 25–50 years. Since then, the standard of care for cardiovascular disease and HF prevention has evolved substantially, including widespread use of statins, evidence-based neurohormonal blockade, and contemporary glucose-lowering agents with cardioprotective effects. These advances may influence both baseline HF risk and the incremental benefit of fibrates today. Therefore, caution is warranted in extrapolating historic trial results directly to present-day clinical practice.

Third, limited statistical power in these studies might have hindered the detection of differences in relatively infrequent outcomes, such as HF hospitalisation or HF-related mortality. However, a meta-analysis of RCTs investigating fibrates as a class of lipid-lowering agents has failed to significantly reduce HF hospitalisation, highlighting inconsistencies in the clinical data.

The post hoc analysis of the ACCORD-Lipid trial provides the strongest clinical indication of a potential benefit, with fenofibrate reducing HF hospitalisation and cardiovascular death, particularly in patients receiving standard glucose control [17]. This finding raises important questions regarding the interplay between metabolic regulation and HF risk modulation by PPARα agonists. However, the absence of benefit in individuals undergoing intensive glycaemic control suggests that the therapeutic effects of fenofibrate may be context-dependent and potentially influenced by metabolic status, disease stage, or concurrent pharmacotherapy. Additionally, significant heterogeneity in fibrate trials underscores the need for targeted investigation into specific patient subgroups that may benefit most from PPARα activation. Evidence indicates that PPARα variants influence fenofibrate efficacy [95% CI 28–66]. No consistent associations with toxicity have been demonstrated; thus, genotype-guided safety selection is not yet warranted. However, pre-specifying PPARα variants for stratified randomisation in future trials may be considered.

Fenofibrate, the most extensively studied fibrate in large RCTs (FIELD, ACCORD-Lipid), has demonstrated a generally acceptable safety profile, with only a reversible rise in serum creatinine and low rates of hepatic, gallstone, or muscle complications when appropriately monitored. These findings suggest that, with clear eligibility criteria, dose adjustments in chronic kidney disease (CKD,) and structured renal, hepatic, and muscle safety monitoring, a dedicated HF-focused RCT of PPARα agonists would be both feasible and clinically justified.

Given the growing recognition of metabolic dysfunction leading to hypertrophy and fibrosis as key drivers of HF pathophysiology, particularly in diabetic cardiomyopathy and HFpEF, there is a strong rationale for prospective trials designed to evaluate PPARα agonists in HF populations. Future studies should prioritise assessing surrogate functional endpoints, such as myocardial energetics, cardiac remodelling, and exercise capacity, alongside traditional HF outcomes to fully elucidate these agents’ therapeutic potential. Addressing these knowledge gaps through well-designed clinical trials will determine whether PPARα agonists can be effectively integrated into HF management strategies.

## 3. Conclusions

Preclinical studies suggest that fenofibrate and other PPARα agonists may reduce the risk of HF through multiple mechanisms, including preservation of myocardial energy metabolism, attenuation of inflammation and oxidative stress, reduction of fibrosis, and protection against ischaemic injury. These findings underscore the multifactorial role of PPARα activation in counteracting key pathophysiological processes implicated in HF progression.

Despite these promising mechanistic insights, clinical translation remains limited. Existing cardiovascular outcome trials were not designed with HF as a primary endpoint and are confounded by heterogeneity in glycaemic control, patient populations, and outcome measures. Future RCTs should evaluate the safety and efficacy of PPARα-targeted therapies, such as fenofibrate, as adjunctive treatments in HF, particularly in patients with metabolic comorbidities, including T2DM and HFpEF.

## Figures and Tables

**Figure 1 biomedicines-13-02080-f001:**
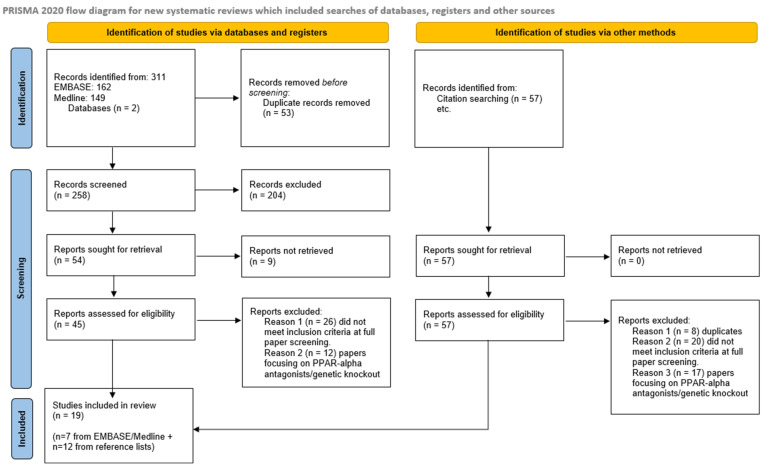
Study flow. PRISMA chart of preclinical studies included in the analysis.

**Figure 2 biomedicines-13-02080-f002:**
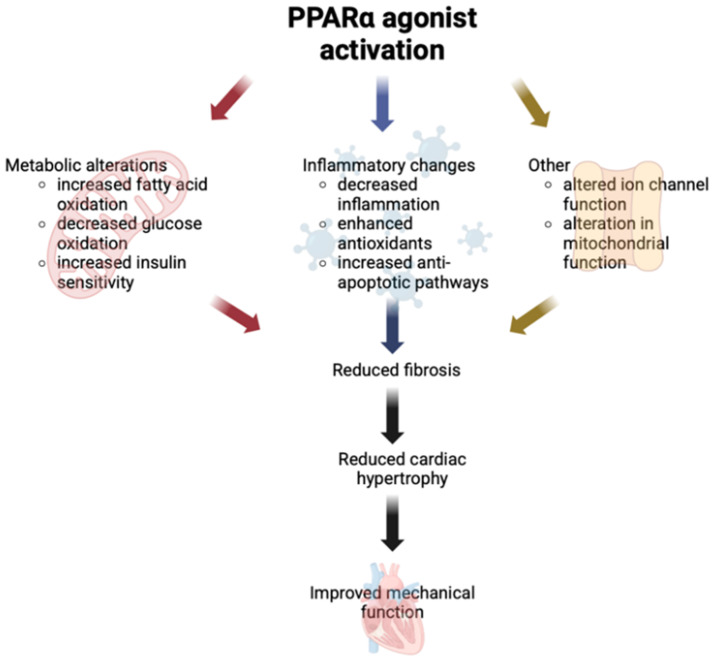
Overview of the molecular pathways through which PPARα activation may enhance cardiac mechanical function. Activation of peroxisome proliferator-activated receptor alpha (PPARα) promotes transcriptional upregulation of genes involved in fatty acid uptake and oxidation, mitochondrial biogenesis, and energy metabolism. These effects contribute to improved myocardial energetics, reduced lipotoxicity, attenuation of oxidative stress and inflammation, and preservation of sarcomeric integrity. Collectively, these mechanisms support enhanced contractile performance and diastolic function in metabolically stressed or failing myocardium.

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
