# Peer review of "Targeting Cardiac Metabolism in Heart Failure with PPARα Agonists: A Review of Preclinical and Clinical Evidence"

_biomedicines, 2025, doi:10.3390/biomedicines13092080_

Round 1

Reviewer 1 Report

Comments and Suggestions for Authors

This is a good review on the therapeutic potential of PPARa for HF. A few suggestions to the authors to add to the review:

  1. Consider discussing the weight of evidence for each study or group of studies.
  2. Mention potential side effects based on available evidence. Speculate on methods/approaches (e.g., adjuvants, combination therapy... etc.) to mitigate these undesirable side effects. Equally important, what clinical or lab parameters must be monitored in clinical trials?
  3. Pharmacogenetics are important. How might polymorphisms in PPARa influence response to therapy in terms of efficacy and toxicity?
  4. Is PPARa activity influenced by different HF phenotypes or metabolic states? How does this impact its therapeutic potential? Perhaps a biomarker-based phenotyping approach would be beneficial? 
  5. The authors must mention underlying issues contributing to the failure of fibrates in cardiovascular trials such as design limitations or patient selection. Could it also simply be a true lack of efficacy?
  6. Based on their analysis, do the authors find it reasonable to encourage interim strategies such as off-label use in highly selected patients? Discuss.
  7. Guide future in vitro studies if recommended. What cellular models, compounds, and pathways have high priority for studying?

Reviewer 2 Report

Comments and Suggestions for Authors

On line 129, the authors state, " Quantitative meta-analysis was not performed due to methodological and population differences among the included studies"

Can the authors expand upon what these methodological and population differences were and provide an example as to why quantitative approaches would not be appropriate, if not even for a subset of the data included?

Line 134 states, "A selection of RCTs was done separately."

Can the authors expand on the selection process of RCTs and include a bit more detailing these?

For the trials in Table 3, please consider listing the sample size in each. Also, perhaps worthwhile to bring up in the discussion, consider the impact of the trials being conducted some time ago, for example, the most recent in 2010 which is 15 years ago and the most distant over 50 years ago. Please comment on how one should interpret the results therein, and perhaps how the potential trt effects might be different today when considering current SoC

For the tables listing other studies, including preclinical/clinical/etc., please add a column indicating the sample size

Round 2

Reviewer 1 Report

Comments and Suggestions for Authors

The authors have adequately addressed my comments.